# HAPPY MAMA Project (PART 1). Assessing the Reliability of the Italian Karitane Parenting Confidence Scale (KPCS-IT) and Parental Stress Scale (PSS-IT): A Cross-Sectional Study among Mothers Who Gave Birth in the Last 12 Months

**DOI:** 10.3390/ijerph18084066

**Published:** 2021-04-12

**Authors:** Alice Mannocci, Azzurra Massimi, Franca Scaglietta, Sara Ciavardini, Michela Scollo, Claudia Scaglione, Giuseppe La Torre

**Affiliations:** 1Faculty of Economics, Universitas Mercatorum, 00186 Rome, Italy; 2Department of Public Health and Infectious Diseases, Sapienza University of Rome, 00185 Rome, Italy; azzurra.massimi@uniroma1.it (A.M.); f.scaglietta@gmail.com (F.S.); ciavardini.1695615@studenti.uniroma1.it (S.C.); scollo.1830613@studenti.uniroma1.it (M.S.); scaglione.1634888@studenti.uniroma1.it (C.S.); giuseppe.latorre@uniroma1.it (G.L.T.)

**Keywords:** distress, self-efficacy, maternal confidence, maternal wellbeing, post-partum

## Abstract

The purposes of this study were: (1) to adapt two validated questionnaires used to evaluate maternal confidence (KPCS-IT) and maternal stress (PSS-IT) to the Italian context, in order to (2) measure the stress level and the self-efficacy in an Italian sample of mothers. The validation process has provided the construction of an online questionnaire. It was administered on a convenience mothers sample with at least a child aged 0–12 months, twice (T0 and T1) with a two day interval. Assessment of instrument stability over time was estimated by applying test–retest reliability between T0 and T1, and the Cronbach’s alpha coefficient. A cross-sectional study was carried out to assess the second aim. Italian mothers with at least one child living at home aged between 0–12 months were recruited. Statistical reliability methods were applied to assess the internal validity of the two questionnaires. PSS-IT was analyzed using univariate and multivariate statistical analyses in order to study the association between KPCS-IT, demographic and maternal characteristics. Statistical significance was established as *p* < 0.05. The Cronbach’s alpha reported a good level of internal consistency of the questionnaires: PSS-IT alpha = 0.862; KPCS-IT alpha = 0.801. 32% of the mothers declared low maternal confidence and the mean value of PSS-IT was 35.4 (SD = 8). The significant inverse correlation was found between the PSS-IT and the KPCS-IT (coeff = −0.353; *p* < 0.001): this means that a high level of perceived self-efficacy reduces the maternal stress level. The study identifies that interventions on maternal confidence can be useful to support mothers in the first months after delivery in order to prevent stress risk: the perceived self-efficacy is as a modifiable factor and the results of the study indicate that it significantly reduces the PSS-IT and EPDS scores. In future, more field trials are necessary in order to assess the realistic and feasible interventions on maternal confidence and competence to prevent maternal distress.

## 1. Introduction

Post-partum is a challenging period for mothers, characterized by many deep changes and many developmental stages in the family life cycle [1,2]. During a child’s first year, mothers often feel overwhelmed by the new situation and need help and support from their partners, their network and health professionals [3]. However, this need for help and guidance is often under-reported, underestimated [4] and usually unmet [5,6,7].

It is fundamental for health professionals involved in primary care who work to promote well-being among new families (i.e., family health nurses and midwives) to identify mothers with low confidence, low mood and high stress in the post-partum period [8,9]. The mother’s well-being, in fact, may impact on the mother–infant relationship and on the infant’s future health, especially during the first year [9].

Several studies have identified low maternal confidence [5], symptoms of depression [10] and parental stress [11] as factors negatively related to the well-being and development of the dyads (mother and infant).

However, only a few standardized screening tools have been evaluated in community settings to assess maternal mood, parental stress and maternal confidence [8].

There are tools concerning the measures used in the literature for assessing maternal stress, confidence and depression. They can be categorized according to the age of the baby.

The Parental Stress Scale (PSS), for example, represents a questionnaire published by Berry et al. [12], that aims to measure parental stress in mothers and fathers with 0–12 months babies.

Confidence may be defined as the amount of beliefs or judgments that a parent holds of his/her capabilities to organize and execute a set of tasks related to parenting a child [13]. It was also defined as the Perception of Parental Self-Efficacy (PPSE). In the literature, there are several instruments to measure the PPSE, such as the Karitane Parenting Confidence Scale (KPCS) [14,15], which can be used for parents of infants aged up to 12 months.

The outcome of post-partum depression, unlike others, is widely investigated. The Edinburgh Postnatal Depression Scale (EPDS) is one of the main tools developed to assess maternal mood [16,17] and is actually validated in the Italian context [16,18].

At present, validated screening tools assessing maternal confidence and stress are lacking in the Italian context. Consequently, there are no Italian studies measuring the prevalence of the distress amongst new Italian mothers.

The first objective of this study is to adapt two validated questionnaires used to evaluate maternal confidence (Karitane Parenting Confidence Scale ITalian version, KPCS-IT) and maternal stress (Parental Stress Scale ITalian version, PSS-IT) to the Italian context, and to evaluate their reliability, their stability over time and their internal consistency [19].

The second and main aim of the study is to measure the stress level, the self-efficacy and the depression risk in an Italian sample of mothers with at least one child under the age of 12 months.

## 2. Material and Methods

The present study was divided in two main sections: the validation of the instruments and a cross-sectional study (the “HAPPY MAMA web survey”).

The University Sapienza of Rome Ethics Committees approved the study (protocol number 826/19RIF.CE: 5559).

### 2.1. Sample

Mothers with at least a child aged 0–12 months were involved in the validation study and in the web-survey.

### 2.2. Validation of the Instruments

An online questionnaire was used to measure (a) the stress level, (b) self-confidence and (c) risk of depression.

The first two outcomes, a–b, were measured creating an Italian version of the KPCS (KPCS-IT) and PSS (PSS-IT). The risk of depression was measured using the Italian version of EPDS validated by Benvenuti et al. [18].

The questionnaires were chosen on the basis of a search on PubMed using the following search algorithm: self-efficacy OR confidence OR stress OR depression AND (mother OR new–mother OR parental) AND (questionnaire OR tool OR score OR scale OR measure). The questionnaires founded were be evaluated in a consensus meeting. It involved the Happy MAMA research group with two psychology and cognitive science experts.

The KPCS [14] is a tool designed to assess the perceived parental self-efficacy defined as “beliefs or judgment a parent holds of their capabilities to organize and execute asset of tasks related to parenting a child” [13]. The 15-item scale grounded in the self-efficacy theory [20] returns a score that ranges from 0 to 45. The cut-off score for KPCS was determined according to the Črnčec et al. [14]: parents scoring ≤39 be experiencing low levels of parenting confidence. The transformation of continuous variable to a binary outcome was applied [14].

The PSS is an 18-item questionnaire, and each item is rated on a scale from 1 to 5 [12]. The PSS final score ranged from 18 to 90: a higher score indicates high level of parental stress. A clinical cut-off point for the PSS was not recommended [12].

The EPDS is a self-report screening measure to detect symptoms of postpartum depression. It is a 10-item questionnaire where each item is rated on a scale from 0 to 3 [21]. The clinical cut-off points in different language versions of the EPDS range from 7 to 14 [22]. Internationally the most commonly used clinical cut-off is an EPDS score ≥12: scores >12 on the EPDS are correlated with a diagnosis of major risk of depressive disorder (MDD) [23]. The transformation of continuous outcome to a binary outcome was applied [23].

The PSS-IT and KPCS-IT validations have foreseen the following activities [24]:three independent researchers translated the English version of KPCS and PSS in Italian language, and then a consensus version was realized;the draft of the Italian version was back-translated in English by an interpreter in order to estimate the compliance with the original version and subsequently it was reviewed according to the translation;a telephone survey was conducted on an opportunistic sample of 30 mothers with at least a child aged 0–12 months. It was carried-out in order to obtain feedback on the level of the items comprehension;a second phone survey was conducted in order to assess the stability and to verify the level of comprehension of the new “final” version: the tool was administered twice over a period of two days to the same group of individuals;the final version was transformed in an online questionnaire using Google Form. A convenience mothers sample (called “validation sample”) with at least a child aged 0–12 months was involved to complete the same questionnaire twice (T0 and T1) with a two day interval (a “Whatsapp” message or an e-mail containing the link to the questionnaire were used as a reminder). Informative notes, aims and details of the study were reported at the beginning of the questionnaire.

These final versions were called PSS-IT and KPCS-IT.

### 2.3. HAPPY MAMA Web Survey

#### 2.3.1. Design

The Strengthening the Reporting of Observational Studies in Epidemiology (STROBE) statement for cross-sectional study was followed to perform the HAPPY MAMA web survey [25,26].

#### 2.3.2. Measures

A questionnaire including the PSS-IT, the KPCT-IT, the EPDS and a section with demographic variables (age, civil status, number of sons, region, city, date of the birth, occupation, vaginal or cesarean delivery) were used for data collection of the second aim of the study: “HAPPY MAMA web survey”.

The data collection phase was performed in May–June 2019.

#### 2.3.3. Data Collection Strategy

The dissemination of the questionnaire link was made through Facebook groups. Italian Facebook groups with a topic on mother, newborn, pregnancy and post-partum were selected. The link was advertised at least once a week during 40 days.

The “HAPPY MAMA project” Facebook page was created and the questionnaire link was posted.

### 2.4. Statistical Analysis

The statistical analysis presented in this paragraph was divided in two sections: one dedicated to the validation of the instruments and the second one to the HAPPY MAMA web-based survey. The validation of the instruments includes a reliability analysis. It was applied on PSS-IT KPCS-IT questionnaires.

The following aspects were considered:

1.Assessing of instrument stability over time: test-retest reliability was estimated between T0 and T1, it was calculated with Intra-class Correlation Coefficient (ICC) with absolute agreement and two-way mixed model. The ICC was estimated using the data of “validation sample”. Test-retest reliability coefficients vary between 0 and 1 and the interpretation by Streiner et al. [27] was considered:1: perfect reliability;≥0.9: excellent reliability;≥0.8 < 0.9: good reliability;≥0.7 < 0.8: acceptable reliability;≥0.6 < 0.7: questionable reliability;≥0.5 < 0.6: poor reliability;<0.5: unacceptable reliability;0: no reliability.
2.Correlation: Spearman’s coefficient was computed between T0 and T1 of PSS-IT e KPCS-IT scores considering the “validation sample”.3.Assessing internal consistency: Cronbach’s alpha coefficient of each questionnaire was applied using a random selection of a sub-group from the sample of the web-survey.

The analysis of the HAPPY MAMA web-based survey included a descriptive, univariate, bivariate and multivariate analysis.

The outcomes considered in the HAPPY MAMA web-based survey were the score obtained by PSS-IT and Benvenuti’s EPDS version [18]. The KPCS-IT score was used as a dependent variable and risk factor.

The descriptive statistics of background variables and outcomes were realized using means, SDs for continuous variables and frequencies with percentages for qualitative ones.

The univariate analysis was realized in order to assess the possible association with PSS-IT versus age, having sons, delivery characteristics, months after delivery and KPCS and EPDS scores. The central limit theorem ensures that parametric tests can be used with large samples (n  >  30), even if the hypotheses of normality are violated [28,29]. Therefore, the T-Student and ANOVA parametric tests were used for the comparison of parental stress between groups of subjects.

ANOVA with a post-hoc Bonferroni test was used to determine the differences among the four quarters (Months after delivery).

The Pearson’s correlation coefficients (r) were calculated to investigate how strongly the three measurements (KPCS-IT, EPDS and PSS-IT) were internally related.

To determine predictors of the PSS-IT score a multivariate linear regression model was performed. The inclusion of any covariate in the model was decided on the basis of the univariate analysis (*p* value ≤ 0.25). The fit of the data into the model was tested using the R^2^.

Stepwise with backward elimination of non-significant variables (probability to entry *p*  <  0.05) was subsequently used to generate a minimal model.

The level of significance was set at *p* < 0.05 for all analysis.

Statistical analysis was performed using the Statistical Package for Social Sciences (SPSS version 25, IBM Corporation, Armonk, NY, USA).

## 3. Results

### 3.1. Validation of the Instruments

The phone survey used to perform the Italian translation was conducted on 29 mothers (see point “d” of “Validation of the instruments” in the previous subparagraph). Among those 26 filled out the questionnaire twice and reported the comprehensibility problems of the items.

The “validation sample” included 25 women, of which 22 filled out the questionnaire twice (see point “e” of “Validation of the instruments” in the previous subparagraph).

#### 3.1.1. Translation and Level of the Items Comprehension

The first phone survey for the feedback on the level of the questions comprehension, showed the need of changes in some items (Appendix A):In the KPCS questionnaire followed items were reviews: ○item 1: “I am confident about feeding my baby…” literally translated into Italian “mi sento fiduciosa quando nutro il mio bambino…” it was then translated into “Mi sento serena quando do da mangiare al mio bambino” with a more familiar tone;○item 3: “I am confident about helping my baby to establish a good sleep routine” literally translated into Italian “Mi sento fiduciosa di aiutare il mio bambino a stabilire un buon ritmo del sonno” it was then translated into “Mi sento in grado di aiutare il mio bambino a stabilire un buon ritmo del sonno” with a more familiar tone;○item 7: “I am confident about playing with my baby” literally translated into Italian “Mi sento fiduciosa quando gioco con il mio bambino”, it was then translated into “Mi sento tranquilla quando gioco con il mio bambino” with a more familiar tone;
In the PSS questionnaire:○item 4: “I sometimes worry whether I am doing enough for my child/ren”, literally translated into Italian “A volte mi preoccupo se sto facendo abbastanza per mio/miei figlio/i” was then translated into “Mi capita di preoccuparmi di non riuscire a fare abbastanza per mio/miei figlio/i” with a more familiar tone.


#### 3.1.2. Reliability of the Instruments

The second phone survey was given to the “validation sample” and it permitted a test-retest analysis. The ICC was 0.957 between T0 and T1 for PSS-IT score and 0.999 T1 for KPCT-IT. Both coefficients indicated excellent agreement between the measures over time.

Furthermore, the Spearman’s coefficient for PSS-IT between T0 and T1 is r = 0.839, *p* < 0.001, and for KPCS-IT is r = 0.999, *p* < 0.001.

The Cronbach’s alpha reported a good level of internal consistency for both PSS-IT (alpha = 0.862) and KPCS-IT (alpha = 0.801). The variation of the alpha value is shown in Table 1 if one item is deleted from the questionnaire. The alpha value was not increased by the removal of items in both questionnaires, except for item 2 and 4 in PSS-IT, but the increase was lower than 0.01.

### 3.2. Web Survey Analysis: Descriptive, Univariate and Bivariate Analysis

The HAPPY MAMA web survey involved 49 Facebook groups, of which 36 published the questionnaire link (Appendix A, see Appendix A). One-thousand and eighty-seven questionnaires were collected. Eight-hundred and seventy-six were valid, the remaining 211 came from pregnant women or mothers with children out of target age. The mean age of women involved in the web survey was 33.8 with SD = 4.5 years. Thirty-one percent of births occur by C-section. Ninety-eight percent had an occupation. The geographical distribution of the mothers was: 28% in the Norther Regions, 47% in the Central and 25% in Southern.

The description of the dichotomized and continuous scores of the KPCS-IT and EPDS (according to the categorization cited in “Material and Methods”) are presented in Table 2. The mean values of the scores were: PSS-IT 35.4 (SD = 8.9, CI95%:34.7–36.1); KPCS-IT 36.8 (SD = 5.0, CI95%:36.4–37.2) and EPDS 9.7 (SD = 5.2, CI95%:9.3–10.0).

The univariate analysis was shown in Table 3. There were significant associations between the high PSS-IT score and working women (*p* = 0.016), time elapsed after the delivery (*p* = 0.006), women with two or more children (*p* < 0.001), women with a low level of parenting confidence (*p* < 0.001) and women with a high risk of depression (*p* < 0.001).

The Pearson’s correlation analysis showed significant direct correlations between the PSS-IT versus EPDS (r = 0.595, *p* < 0.001), number of children (r = 0.08, *p* = 0.40) and the number of days after the birth (r = 0.134, *p* < 0.001). An inverse significant correlation was found with KPCS-IT (r = −0.577, *p* < 0.001) as it is illustrated in Figure 1.

Multivariate analysis is reported in Table 4. The findings indicated that the covariates significantly associated to the high PSS-IT level were: days from delivery (coeff = 0.05; *p* < 0.001) and risk of depression (coeff = 0.378; *p* < 0.001); while the covariates significantly associated to the low PSS-IT level were having only one child (coeff = −0.135; *p* < 0.001) and a high self-efficacy score (coeff = −0.353; *p* < 0.001). In the regression model the goodness-of-fit for PSS-IT was R^2^ = 0.455 (dependent variables explain 45.5% of the variability).

## 4. Discussion

The present study supports the validity and the reliability of the PSS-IT and KPCS-IT.

Analyses revealed that the results of both times (T0 and T1) are comparable, thus suggesting the stability of the scale characteristics. Therefore, as parental stress is concerned the PSS-IT has certain advantages: it is easily understandable for mothers and it is brief and easy to administer and score.

According to the literature, the study confirms that the KPCS-IT has easy administration, compilation and scoring [14]. Moreover, the reliability of KPCS-IT and PSS-IT measures of this study are consistent with previous findings: PSS-IT alpha = 0.84% and for KPCS-IT alpha = 0.74% [8].

The second part of the research focused on the cross-sectional study, suggested that the stress level (PSS-IT) is significantly inversely correlated to the self-efficacy (KPCS-IT). This outcome is in agreement with the study’s hypothesis. Additionally, a directly significant correlation between the PSS-IT and the EPDS is confirmed.

The correlations of maternal mood, maternal confidence and parental stress are consistent with the previous findings, even when using alternative measures for assessing the parental self-efficacy [8,30]. The obtained mean value of PSS-IT was in agreement with the results of Berry et al. [12] Indeed, their study showed the mean of PSS-IT score in two different groups: in the clinical group it was 43.2 (SD = 9.1; n = 51) and in the control group was 37.1 (n = 116, SD = 8.1; CI95%:35.6–38.6).

Furthermore, the findings of this study underline a gradual significant increase in the parental stress during the 12 month postpartum, and this is in agreement with the results of previous publications [8,31,32].

This study had several limitations. Concerning the first part of the study, “validation and reliability”, the test-retest analysis could be not robust, because the guidelines recommended using a sample size as large as possible, but given the variation in the types of used questionnaire, there are no absolute rules for the sample size needed to validate a questionnaire [24]. Moreover, although the sample size was comparable to numerous other test-retest reliability studies [33], small sample sizes have the limit of creating some instability in the alpha coefficients and results must be interpreted with caution.

Concerning the limitations of the second part of the study, a peculiar limitation of the Web survey is due to the cross-sectional design, because the exposure (maternal self-efficacy perception) and outcome (maternal stress level) are simultaneously assessed, there is generally no evidence of a temporal relationship between exposure and outcome.

Secondly, the external validity of the study may be affected by the exclusive presence of Facebook social network female users. It may have influenced the individual characteristics. It is nonetheless true that female social network users in 2018 represent the 66% of female population between 16–74 of age according to EUROSTAT and ISTAT data [34,35].

Another limit of the web-survey was the low sample size. In Italy, the births are about 435,000 for a year [36]. The size of the sample studied was n = 1087, which means that about 0.25% of the population are new mothers.

A possible selection bias could be included in the results. The women with a premature delivery, or infants with some sequelae after the birth or a major complication in the delivery could have participated more than women with no delivery or infant complications. This aspect could underestimate the stress levels and this could affect the research results.

Furthermore, the study did not take into account if there were external supports such as child-care interventions. This aspect can reduce significantly the stress-level.

Moreover the coefficient of multiple determination for regression model, R^2^, indicated that the strength of the relationship between the model and the PSS-IT was low. Approximately half of the observed variation of PSS-IT can be explained by the model’s inputs.

## 5. Conclusions

In conclusion, about one third of the mothers involved in the study declared low maternal confidence and 31% had a high risk of depression. The mean value of parental stress score was in agreement with the literature.

Furthermore, the study identifies potential areas of social support and practice: interventions on maternal confidence may be needed to support mothers in the first months following delivery in order to prevent stress risk: the perceived self-efficacy is a modifiable factor and the results of the present study indicate that it reduces at significant level the parental stress and depression risk. In the future, more research in this area is required and, in particular, more field trials are necessary in order to assess realistic and feasible interventions on maternal confidence and competence to prevent maternal distress.

## Figures and Tables

**Figure 1 ijerph-18-04066-f001:**
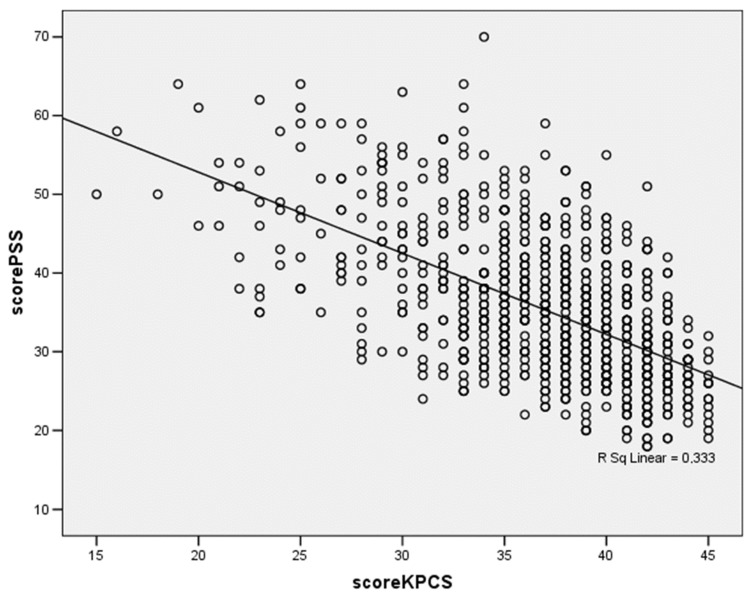
Bivariate analysis: scatter-plot between the PSS-IT and KPC-IT scores.

**Table 1 ijerph-18-04066-t001:** Cronbach’s alpha of KPCS-IT, PSS-IT scores.

Questionnaire	Item	Cronbach’s Alpha if Item Deleted
	1	0.850
	2	**0.863**
	3	0.859
	4	**0.863**
	5	0.857
	6	0.856
	7	0.861
	8	0.857
PSS-IT	9	0.852
	10	0.853
	11	0.858
	12	0.851
	13	0.855
	14	0.856
	15	0.847
	16	0.849
	17	0.852
	18	0.859
	**Pooled Cronbach’s alpha**	**0.862**
	1	0.792
	2	0.786
	3	0.793
	4	0.783
	5	0.787
	6	0.787
	7	0.787
KPCS-IT	8	0.792
	9	0.805
.	10	0.783
	11	0.783
	12	0.791
	13	0.781
	14	0.791
	15	0.810
	**Pooled Cronbach’s alpha**	**0.801**

**Table 2 ijerph-18-04066-t002:** Descriptive statistics of HAPPY MAMA web-based survey’ sample.

Variables			
Qualitative		n	%
Cesarean delivery	No	270	69
	Yes	606	31
Gestational age (weeks)	≥38	782	89
	<38	94	11
Lives with infant’s father	Yes	856	98
	No	12	1
	No answer	8	1
Geographical area where she lives	North	248	28
	Center	410	47
	South	218	25
Months after delivery (quarters)	1st	192	22
	2nd	251	29
	3rd	191	22
	4th	242	27
Number of sons	1	572	65
	>1	304	35
Age groups (years)	≤31	269	32
	32–35	281	32
	≥36	296	34
Employed/student or housewife	Yes	777	89
	No	99	11
KPCS-IT ^a^ (perception of self-efficacy)	Yes	591	68
	No	283	32
EPDS ^a^ (presence risk of depression)	Low	603	69
	High	273	31
**Quantitative**		**Mean**	**SD**
PSS-IT score		35.4	8.9
KPCS-IT score		36.8	5.0
EPDS score		9.7	5.2

^a^ Dichotomous variable. The cut-off point is defined according to the literature (see “Material and Methods” paragraph).

**Table 3 ijerph-18-04066-t003:** Univariate analysis of PSS-IT score versus the variables studied.

Variables		PSS-IT Score			Test
		Mean	SD	*p*	
Cesarean delivery	Yes	35.62	9.04	0.774	T-student
	No	35.44	8.96		
Gestational age (weeks)	≥38	35.2	8.2	0.688	T-student
	<38	35.5	9.2		
Employed	Yes	35.2	8.8	0.016	T-student
	No	37.5	9.6		
Geographical area where she lives	North	35.57	8.45	0.358	Anova
	Center	35.09	9.26		
	South	36.16	9.01		
Months after delivery (quarters)	1st	34.84	8.52	**0.006 ***	ANOVA *
	2nd	34.59	8.81		
	3rd	35.18	8.96		
	4th	37.20	9.33		
Number of sons	1	34.71	9.16	**<0.001**	T-student
	>1	36.97	8.44		
Age groups (years)	≤31	34.71	8.97	0.178	Anova
	32–35	35.82	8.97		
	≥36	36.01	8.84		
KPCS-IT ^a^	Yes	30.05	6.58	**<0.001**	Anova
	No	38.13	8.8		
EPDS ^a^	Low	32.45	6.95	**<0.001**	Anova
	High	42.22	9.30		

Bold: The mean difference is significant at the <0.05 level. a. Dichotomous variable. The cut-off point is defined according to the literature (see “Material and Methods” paragraph). * The Bonferroni’s post-hoc analysis to assess the difference between the quarters (*p*-value was set at *p* < 0.05/4 = 0.0125).
**(I) Quarter****(J) Quarter****Mean Difference (I–J)****Bonferroni’s Test *p* Value****95% CI****Lower****Upper**120.250.99−2.012.51
3−0.330.99−2.752.08
4−2.350.04−4.64−0.0721−0.250.99−2.512.01
3−0.580.99−2.851.68
4−2.60**0.01****−4.73****−0.48**310.330.99−2.082.75
20.580.99−1.682.85
4−2.020.12−4.310.26412.350.040.074.64
22.60**0.01****0.48****4.73**
32.020.12−0.264.31

**Table 4 ijerph-18-04066-t004:** Multivariate analysis: linear regression model of PSS-IT score.

Covariates		PSS-IT Score	
		Coefficient	*p*
Employed	Yes	−0.026	0.320
	No *		
Age (years)		0.050	0.053
Days from delivery		0.095	**<0.001**
Number of sons	1	−0.135	**<0.001**
	>1 *		
KPCS-IT score		−0.353	**<0.001**
EPDS score		0.378	**<0.001**
Goodness-of-fit: R^2^		0.455	

* Reference group Bold: *p*-value < 0.05.

## Data Availability

Not applicable.

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
