# Peer review of "HAPPY MAMA Project (PART 1). Assessing the Reliability of the Italian Karitane Parenting Confidence Scale (KPCS-IT) and Parental Stress Scale (PSS-IT): A Cross-Sectional Study among Mothers Who Gave Birth in the Last 12 Months"

_ijerph, 2021, doi:10.3390/ijerph18084066_

Round 1

Reviewer 1 Report

I think the topic and writing is good. However, is the sample size too small?

Author Response

I think the topic and writing is good. However, is the sample size too small?

Thank for your question.  In Italy the births are about 435.000 for year (source: Italian Minister of Health  -Rapporto sull’evento nascita in Italia (CeDAP) – year  2018). About 7% of these have problems (low Apgar score, low weight, genetic diseases, malformations and so on). Our sample N=1087 represent about 0.27% of the population and we have underlined this limit of the study according to your comment.

Reviewer 2 Report

This research studies how the level of maternal confidence directly influences the stress experienced by parents and postpartum depression. It is a very nice and interesting subject.

For this, it will be used in 3 measuring instruments: KPCS, PSS and PSS.  KPCS and PSS were not previously validated in Italian. So the first part of the study consisted of validating these two tests for the Italian population. The second part of the research consisted of conducting a descriptive study to see how PSS and PDS correlate with KPCS.

However, I would like to make some comments for the sole purpose of improving the quality of the manuscript.

In the material and method section, it is explained that the ANOVA test was used to relate the continuous variables. (line 176). I think it would be convenient to indicate whether a normality test has previously been performed to relate the quantitative variables, which may justify the use of ANOVA.

In the results section, when the characteristics of the sample are explained, I believe that the inclusion and exclusion criteria should be clearly stated. 

I think that the fact that the child is less than a year old is not enough. I think that if the delivery was premature, the baby had some sequelae after the birth or there was some major complication in the delivery, the stress levels of the parents will be very high and this could affect the research results.

In line 264 it says that the multivariate analysis is represented in table 4. In the manuscript I cannot find table 4. I think it was not put,
I would like to see the regression model.

The introduction, discussion and conclusions section seems correct to me. The validation process of the questionnaires also seems adequate to me.

Kind regards.

Author Response

In the material and method section, it is explained that the ANOVA test was used to relate the continuous variables. (line 176). I think it would be convenient to indicate whether a normality test has previously been performed to relate the quantitative variables, which may justify the use of ANOVA.

Thanks for your comment. We have reported the explanation of test used in Methods  as below.

“The central limit theorem ensures that parametric tests can be used with large samples (n > 30), even if the hypotheses of normality are violated [Rao, 1998]. The t test, analysis of variance (ANOVA), and analysis of covariance (ANCOVA) on ordinal data have been shown to be robust to violations of normality with small samples as well [Heeren, 1987]. Therefore, the T-Student and ANOVA parametric tests were used for the comparison of maternal mood, maternal confidence and parental stress between groups of subjects.”

In the results section, when the characteristics of the sample are explained, I believe that the inclusion and exclusion criteria should be clearly stated. 

Thanks. We have added  the inclusion criteria in "Methods" paragraph as below:

“Mothers with at least a child aged 0-12 months were involved in validation study and in the web-survey.”

I think that the fact that the child is less than a year old is not enough. I think that if the delivery was premature, the baby had some sequelae after the birth or there was some major complication in the delivery, the stress levels of the parents will be very high and this could affect the research results.

The stress level was studied in general in the population of the new-mothers. The tools were designed and validated for all “categories” of the new-mothers.

In the web-survey we know that we have included “outlier” cases, but the aim of study was measure the stress level of the all categories of new-mothers, and no the new-mothers with a "normal" delivery. 

We have consider, however, your comment. It was an interesting question for us. We have decided to included the possibility of selection bias in the “limits of the study", as below:

“A possible selection bias could be include in the results. The women  with a premature  delivery, or  infant  having some sequelae after the birth or major complication in the delivery could have participate more than women with no delivery or infant complications. This aspect could be underestimate the stress levels and this could affect the re-search results.”

In line 264 it says that the multivariate analysis is represented in table 4. In the manuscript I cannot find table 4. I think it was not put,
I would like to see the regression model.

We added the table 4.

The introduction, discussion and conclusions section seems correct to me. The validation process of the questionnaires also seems adequate to me.

Reviewer 3 Report

This paper reports a study which addresses two significant questions regrading the psychological coping capacity of mothers soon after delivery of their children. Firstly the authors address the validity of Italian versions of two well established psychological instruments and secondly they report the results of those questionnaires and a third well known instrument, in their study population. However the first sentence of their "Abstract" refers only to the first research question, although the second purpose of the study is addressed later in the "Introduction". 

The study is well constructed to address the questions they have posed, with recognition that the study population was drawn from a cross-sectional sample. The numbers of subjects are adequate for their statistical analysis and the form of that analysis is appropriate. I found the author's approach to the design of the Italian version of the questionnaires to be thorough and the tests of validity and reliability satisfactory. In my opinion the Italian versions of these questionnaires appear suitable for further use.

The findings regarding the observational element of the study were of interest and offered valuable insights into the coping capacity of these mothers and the opportunities for therapeutic intervention (including prevention). I again found the analysis of this data to be appropriate and the conclusions justified by the data.

With regard to the English used in this paper, I have a number of minor suggestions:

Line 50: Change "settings" to "setting".

Lines 200/201: Change "...valid..." to "...were valid...".

Line 223: Change "...have allowed to access..." to "...permitted a test-retest...".

Line 271: Change "...and the reliability..." to "...and reliability...".

Line 324: Change "self efficacy as a modifiable..." to "...is a modifiable..."

Author Response

This paper reports a study which addresses two significant questions regrading the psychological coping capacity of mothers soon after delivery of their children. Firstly the authors address the validity of Italian versions of two well established psychological instruments and secondly they report the results of those questionnaires and a third well known instrument, in their study population. However the first sentence of their "Abstract" refers only to the first research question, although the second purpose of the study is addressed later in the "Introduction". 

We reviewed the abstract as below: “The purposes of this study were: 1) to adapt two validated questionnaires used to evaluate maternal confidence (KPCS-IT) and maternal stress (PSS-IT) to the Italian context, in order to 2) measure the stress level, the self–efficacy in an Italian sample of mothers”.

The study is well constructed to address the questions they have posed, with recognition that the study population was drawn from a cross-sectional sample. The numbers of subjects are adequate for their statistical analysis and the form of that analysis is appropriate. I found the author's approach to the design of the Italian version of the questionnaires to be thorough and the tests of validity and reliability satisfactory. In my opinion the Italian versions of these questionnaires appear suitable for further use.

The findings regarding the observational element of the study were of interest and offered valuable insights into the coping capacity of these mothers and the opportunities for therapeutic intervention (including prevention). I again found the analysis of this data to be appropriate and the conclusions justified by the data.

With regard to the English used in this paper, I have a number of minor suggestions:

Line 50: Change "settings" to "setting".

Lines 200/201: Change "...valid..." to "...were valid...".

Line 223: Change "...have allowed to access..." to "...permitted a test-retest...".

Line 271: Change "...and the reliability..." to "...and reliability...".

Line 324: Change "self efficacy as a modifiable..." to "...is a modifiable..."

Thanks, for the English suggestions.

Reviewer 4 Report

Thank you for this interesting topic. However, in my view, much more work should be done to improve the scientific quality of the manuscript. In my opinion, it has serious flaws as listed below:

a) weak research motive, i.e. the introduction is quite vague, core concepts are neither well described nor contextualized;

b) no clarity regarding some methodological choices, starting from the selection of instruments;

c) incomplete data;

d) inaccurate conclusions and assumptions are not supported by results.

Furthermore, the use of English is quite poor as well as style and writing. The title is inappropriate and inconsistencies are several throughout the paper.

Author Response

Thank you for this interesting topic. However, in my view, much more work should be done to improve the scientific quality of the manuscript. In my opinion, it has serious flaws as listed below:

  1. weak research motive, i.e. the introduction is quite vague, core concepts are neither well described nor contextualized;

Your comments are harsh criticisms, but we thank for them. You have give us the opportunity to improve the clarity and comprehensibility of the document. I hope that our efforts have improve our work.

So, we have organized in different way the paragraphs and subparagraphs.

We have organized the “Results” according to the “Methods” and the “Aims”.  

All changes were reported using Track Changes Word tool.

  1. no clarity regarding some methodological choices, starting from the selection of instruments;

We have organized in different way the paragraphs and subparagraphs of the “Methods”.

The strategy of instruments was not been included for problems of words amount. Now, we have  included the choice of the tools  in "Methods" paragraph, as below: “The questionnaires were been chosen on the basis of a search on PubMed using the following search algorithm: self-efficacy OR confidence OR stress OR depression AND (mother OR new –mother OR parental) AND (questionnaire OR tool OR score OR scale OR  measure). The questionnaires founded were be analyzed in a consensus meeting. It was organized with the Happy MAMA research group with two psychology and cognitive science experts”.

  1. c) incomplete data;

The data was added, for example  Table4.

  1. inaccurate conclusions and assumptions are not supported by results.

The "Discussion" and the assumptions were reviewed.

Furthermore, the use of English is quite poor as well as style and writing. The title is inappropriate and inconsistencies are several throughout the paper.

The English review performed with a support with a native speaker.

Round 2

Reviewer 2 Report

This research studies how the level of maternal confidence directly influences the stress experienced by parents and postpartum depression. It is a very nice and interesting subject.

For this, it will be used in 3 measuring instruments: KPCS, PSS and PSS.  KPCS and PSS were not previously validated in Italian. So the first part of the study consisted of validating these two tests for the Italian population. The second part of the research consisted of conducting a descriptive study to see how PSS and PDS correlate with KPCS.

This second version of the manuscript is much better than the first. The great effort made by the authors is highly appreciated. They have managed to significantly increase the quality of the manuscript. All recommendations provided by the reviewers have been followed.

Congratulations

Kind regards

Author Response

Many thanks.

Reviewer 4 Report

Thank you for the changes. I really appreciate your efforts. Unfortunately, I do not feel that the paper has sufficiently improved to be considered for publication.  In my view, much more work should be done to improve the scientific quality of the manuscript.

In my opinion, it has serious flaws yet. The introduction is still quite vague, core concepts are neither well described nor contextualized. Focusing on post-partum is a good idea. There is extensive literature on protective and risk factors on new mothers’ well being. This should be seriously considered and summarized with criticism. Moreover, the title seems distant from your focus, which becomes clear only in the introduction.

Overall, in my view, the paper (including the abstract) does not fulfil scientific requirements.  Moreover, to me, it is not easily readable and, at times, confusing (to readers). For example, scores or description of some statistical analyses (p. 4/5 et.) are usually not fully reported, since researchers are assumed to know about them or, alternatively, they can read references. Instead, core concepts cannot be neglected, thus compromising the understanding and the value of the study. Similarly, ethical aspects are usually not included within the paragraph ‘statistical analysis’.

The use of English is not up to a scientific publication as well as there are many stylistic errors such as inappropriate citations, acronyms etc.

Author Response

Thanks for your comments. We hope to have further improved the manuscript following your review.

- The introduction was re-organized and reviewed in order to facilitate the reasoning that has conducted to the purpose of this study.

- The title was reviewed:

HAPPY MAMA Project (PART 1). Assessing the reliability of the ITalian Karitane Parenting Confidence Scale (KPCS-IT) and Parental Stress Scale (PSS-IT): a cross-sectional  study among mothers who gave birth in the last 12 months

We have reviewed the Abstract adding the validation process. See below:

Validation process has provided the construction of an on-line questionnaire. It was administered on a convenience mothers sample with at least a child aged 0-12 months twice (T0 and T1) with a two day interval. Assessing of instrument stability over time was estimated applying test-retest reliability between T0 and T1, and the Cronbach’s alpha coefficient.”

- The scores PSS e KPCS are described in “validation instruments” sub-paragraph of “Methods”.

- The statistical analyses  are  reviewed  as shown in the following:

The univariate analysis was realized in order to assess the possible association with PSS-IT versus age, having sons, delivery characteristics, months after delivery and KPCS and EPDS scores. The central limit theorem ensures that parametric tests can be used with large samples (n > 30), even if the hypotheses of normality are violated [Rao, 1998; Heeren, 1987]. Therefore, the T-Student and ANOVA parametric tests were used for the compari-son of parental stress between groups of subjects.

ANOVA with Post-hoc Bonferroni test was used to determine the differences among the four quarters (Months after delivery).”

- The ethical aspects were reported at the beginning of the paragraph of the “Methods”, after the description of the study design.

- We have reviewed the use of English. The acronyms and citations were checked.